# PROMPT-MII: META-LEARNING INSTRUCTION INDUCTION FOR LLMS

**Emily Xiao, Yixiao Zeng, Ada Chen, Chin-Jou Li, Amanda Bertsch, Graham Neubig**
Language Technologies Institute, Carnegie Mellon University
{emilyx,jackz,adachen,chinjoul,abertsch,gneubig}@cs.cmu.edu

## ABSTRACT

A popular method to adapt large language models (LLMs) to new tasks is in-context learning (ICL), which is effective but incurs high inference costs as context length grows. An alternative approach is to perform *instruction induction*, where we take training examples and reduce them to a compact but descriptive prompt that can achieve performance comparable to ICL over the full training set. We propose PROMPT-MII, a reinforcement learning (RL) based framework to *meta-learn* an instruction induction model that can generate compact instructions on the fly for an arbitrary new dataset. We train on over 3,000 diverse classification datasets from the HuggingFace hub, and evaluate on 90 unseen tasks. PROMPT-MII improves downstream model quality by 4-9 F1 points (10-20% relative), matching ICL performance while requiring 3-13x fewer tokens. Code is available at `https://github.com/millix19/promptmii`. Models and datasets are available at `https://huggingface.co/collections/milli19/promptmii-68f11db8e2a2f775d2f04a1a`.

## 1 INTRODUCTION

One common use pattern for large language models (LLMs) is to adapt them to a specific downstream task. In a supervised adaptation scenario, we are given $n$ labeled demonstrations $S_{\text{train}} = \{(x_k, y_k)\}_{k=1}^n$ and are interested in the problem of how to accurately predict labels for a set of test examples $S_{\text{test}} = \{(x_j, y_j)\}_{j=1}^m$ drawn from the same distribution.

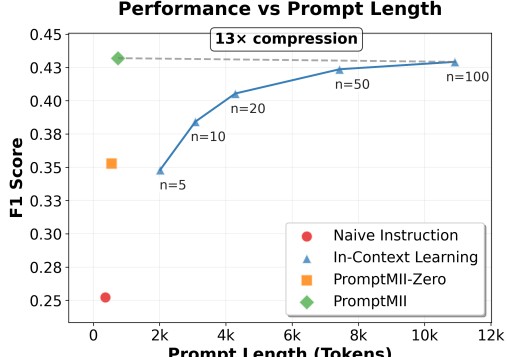

Figure 1: Classification task performance averaged over 90 datasets, using the Llama-3.1-8B-Instruct model. PROMPT-MII achieves performance comparable to ICL while using $13\times$ fewer tokens.

There are multiple typical ways to incorporate the given examples: (1) *Prompting with instructions*, where a natural language task description $I$ is appended to the model prefix, (2) *In-context learning (ICL)*, which directly uses examples in $S_{\text{train}}$ as context during inference, and (3) *Supervised fine-tuning (SFT)*, which performs gradient updates on $S_{\text{train}}$ to condense the information into model parameters. Each method has its advantages. Prompting with instructions is concise and efficient but requires extensive prompt engineering (Sahoo et al., 2024; Schulhoff et al., 2024). ICL achieves highly competitive performance but can be inefficient as the number of examples grows larger (Xiao et al., 2025). SFT is efficient at test time but uses significant compute at training time, requires storage of model weights, and underperforms ICL in many cases (Bertsch et al., 2024).

As a method to bridge the gap between ICL and prompting, there exists *instruction induction*, which takes training data $S_{\text{train}}$ and generates an instruction $I$ that achieves good performance. Representative methods for instruction induction such as APE (Zhou et al., 2022) and GEPA (Agrawal et al., 2025) typically do so through expensive evolutionary search algorithms at test time that generate

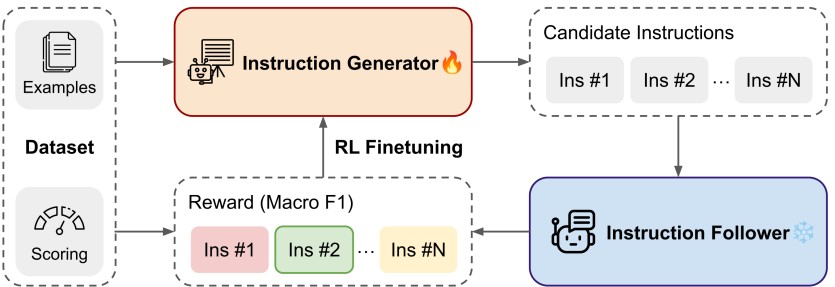

Figure 2: Overview of PROMPT-MII. We train an Instruction Generator's general ability to perform instruction induction. At inference time, given dataset examples of an unseen task, it automatically generates a reusable task instruction in a single pass, which then guides a black-box Instruction Follower model to make predictions.

multiple candidates for prompts and evaluate them to choose a well-performing prompt. This raises the question: *is there a way to perform instruction induction in a way that is both effective and efficient over a wide variety of tasks?*

As an answer to this question, we propose PROMPT-MII, where we frame instruction induction as a meta-learning problem: instead of individually optimizing $I$ for each individual task, we train an instruction induction policy $\pi_\theta$ that can effectively generate instructions in a single pass across diverse task distributions, conditioned only on the in-context examples:

$$I = \pi_\theta(S_{\text{train}}^{(i)}) \tag{1}$$

There are two major advantages to this approach. First, it allows $\pi_\theta$ to share knowledge about how to construct effective prompts across a wide number of datasets, instead of requiring the re-discovery of this knowledge for each dataset. Second, it has significant efficiency benefits – generating an instruction $I$ for a new dataset simply requires a single forward pass through the language model, instead of a costly optimization process.

Experiments demonstrate that PROMPT-MII is highly effective. For instance, in Figure 1 we show how PROMPT-MII can achieve performance comparable to 100-shot ICL while consuming 13x fewer tokens. In the following sections, we discuss the methodological details of PROMPT-MII (§ 2), experimental details (§ 3), and results and analysis (§ 4).

## 2 PROMPT-MII: META-LEARNING INSTRUCTION INDUCTION

The main challenge in developing a method to generate instructions $I$ from a dataset $S_{\text{test}}$ is learning an effective policy $\pi_\theta$ that can generate these instructions in a way that will achieve good test performance. We use reinforcement learning (RL) to train this policy because ground-truth dataset-instruction pairs are often not available for existing public datasets. However, we can evaluate instruction quality through downstream task performance, which serves as a natural reward signal for RL. In this section, we develop our method for meta-learning such a policy, also shown in Figure 2.

### 2.1 TRAINING OBJECTIVE

Let $\mathcal{S} = \{S_1, S_2, \ldots, S_N\}$ be a collection of datasets that we will use in the meta-learning of $\pi_\theta$. For each dataset $S_i$, we sample training examples $S_{\text{train}}^{(i)}$ for instruction generation and test examples $S_{\text{test}}^{(i)}$ for reward computation. We define a meta-prompt template $T(S_{\text{train}}^{(i)})$, which converts the dataset into a prompt to the model, as detailed in § 2.2. Then, $\pi_\theta$ generates an instruction prompted by this meta-prompt, $I \sim \pi_\theta(T(S_{\text{train}}^{(i)}))$.

To assess the quality of the generated instruction, we use a separate frozen language model $\text{LM}_{\text{eval}}$ as the instruction follower. This LM then processes the test set $S_{\text{test}}$ using this instruction, generating

results $\hat{y}_j = \text{LM}_{\text{eval}}(I + \text{"Input: "} + x_j + \text{"Label:"})$. We use a task-dependent evaluation metric over $m$ test examples to assess the model performance $E\left(\{\hat{y}_j\}_{j=1}^m, \{y_j\}_{j=1}^m\right)$. In principle, this metric can range from classification metrics such as accuracy and macro-F1 to generation based metrics such as LLM-as-a-judge, but in this work we focus on classification tasks and use macro-F1 as our target reward metric and $m = 20$ to balance stability and efficiency. To avoid training the model to learn the format requirement that is easily enforced manually, we add this custom format line: `Only return one of these options: {label_names}. Do not output "Label:" or any extra text.` after the generated instruction, before calculating the reward. This constraint is equally added to all baseline methods we compare in the results.

Together, this results in a reward for our generated instruction of

$$R(I, S_{\text{test}}) = E\left(\{\hat{y}_j\}_{j=1}^m, \{y_j\}_{j=1}^m\right) \tag{2}$$

Once we have defined this reward, it can be optimized with an RL algorithm of choice. In this work, we use Group Relative Policy Optimization (GRPO; Shao et al. (2024)) and enhance the algorithm with asymmetric clipping and removal of KL loss, which has been shown to encourage more exploration (Yu et al., 2025). Full details of the RL objective are in § A.1.

## 2.2 META-PROMPT TEMPLATE

One key element of our method is the use of a meta-prompt template $T$ that encourages the LLM to generate instructions with generalizable patterns rather than regurgitating specific examples or simply summarizing the label space.

Meta-prompt design impacting prompt quality is a known phenomenon in automatic prompt optimization (APO) methods (Ding et al., 2025). Our ablation studies in § 4.5 reveal model-dependent preferences, and accordingly, we use model-specific meta-prompts optimized for each model, but fix the same template for training and evaluation of all baselines.

---

**Meta-Prompt Template (Qwen)**

You are designing a clear instruction for a data annotator to classify text inputs into one of these labels: {label_names}
Here are some example inputs and their correct labels: {examples}
Your task is to write a concise instruction that:

- Defines the classification task and clearly explains the meaning of each label.

- Provides general labeling strategies and decision rules so annotators can correctly handle unseen inputs.

- Highlights common pitfalls, tricky edge cases, and misconceptions to reduce labeling errors.

- Keeps the instruction reasonably concise and focused — avoid unnecessary repetition or overly long explanations.

---

Here, {label_names} is a comma-separated list of all of the labels in the classification dataset $S_{\text{train}}$ (e.g., "positive, negative, neutral") and {examples} follows the format: `Text: "example input text here"\nLabel: example_label`. See § A.6 for details.

## 3 EXPERIMENTS

### 3.1 TRAINING DATA PREPARATION

We collected all publicly available text classification datasets from HuggingFace, applied automated filtering, and randomly sampled training examples $S_{\text{train}}^{(i)}$ as described in Appendix § A.2. After filtering, we obtained 3,811 diverse datasets, which were randomly split into 3,430 for training and 381 for validation.

## 3.2 Training Setup

We conducted training using the VERL framework (Sheng et al., 2024) on two model variants: `Llama-3.1-8B-Instruct` and `Qwen-2.5-7B-Instruct`. For each variant, we used the same official model checkpoint for both the instruction generator ($\pi_\theta$) and the instruction follower ($\text{LM}_{\text{eval}}$). While $\text{LM}_{\text{eval}}$ was kept frozen at the official checkpoint, $\pi_\theta$ was updated during training. For training, we used rollout size of $n = 5$, batch size of 64, maximum response length of 1k tokens and maximum prompt length of 4k tokens. Further hyperparameter and system details are provided in Appendix § A.3.

## 3.3 Evaluation Setup

**Data** We randomly select 90 datasets from the validation set for evaluation, which is disjoint from the training set. For each dataset and each $n \in \{5, 10, 20, 50, 100\}$, we sampled $n$ training examples, generated instructions, and applied them to 200 test examples. The context length was limited to 32k tokens. If the $n$ examples exceeded this limit (applicable to ICL and PROMPT-MII), we used the maximum value of $n$ that fit within the context. See § A.2 for further implementation details on dataset selection, and Table 7 for full list of datasets and statistics.

**Baselines** We compared our method against Naive Instruction, In-Context Learning (ICL), PROMPT-MII-Zero (untrained instruction generator), and PROMPT-MII-Zero with larger models (`Llama-3.1-405B-Instruct`, `Qwen-3-235B-Instruct`). We also compare with inference-time search-based prompt optimization methods APE (Zhou et al., 2022) and GEPA (Agrawal et al., 2025). Since our datasets do not provide ground-truth instructions, all baselines we consider are automatic prompt generation methods. Further implementation details are as described in § A.4.

> Prompt Template for Naive Instruction Baseline
>
> Classify the Input. Only return one of these options: {$\text{label}_1$, $\text{label}_2$, ... $\text{label}_n$}. Do not output 'Label:' or any extra text.

> Prompt Template for ICL Baseline
>
> Classify the Input. Only return one of these options: {$\text{label}_1$, $\text{label}_2$, ... $\text{label}_n$}. Do not output 'Label:' or any extra text.
> Input: {Example 1}
> Label: {Label 1}
> ...
> Input: {Test case}
> Label:

**Metrics** Our evaluation metric for task performance is the macro-F1 score, which is consistent with the training reward, and accounts for label imbalance. To assess efficiency, we report the prompt token length, since shorter prompts directly translate to lower inference cost and latency when deployed on LLMs. Additionally, we report win rates (the percentage of datasets where one method outperforms another) and the training curve in § A.5.

## 4 Results

### 4.1 Prompt-MII Successfully Generates Concise and Effective Instructions

RL training consistently improves instruction generation across held-out tasks, providing the first evidence that one-pass instruction induction is a skill learnable by language models. As shown in Figure 6 and Table 1, Llama PROMPT-MII (trained) achieves +9% absolute F1 improvement over PROMPT-MII-Zero (untrained) at n=20 (26% relative gain), while Qwen PROMPT-MII shows +5% absolute improvement (15% relative gain).

Training conducted with limited context length of 4k context length is able to have improvements generalized to 32k context length. Notably, Llama PROMPT-MII using n=20 examples (0.433 F1, 901 tokens) matches ICL performance using n=100 examples (0.430 F1, 11,531 tokens), representing a 12.8× token reduction with no statistical difference in performance, as shown in Table 1 and Figure 3 We also compare the per-dataset win rate between ICL and PROMPT-MII, and find that both prevail in an approximately equal number of tasks (Figure 8, Appendix), PROMPT-MII has a similar win rate to ICL (approximately 50-50). Together, these results suggest that PROMPT-MII is a strong alternative for practitioners to consider.

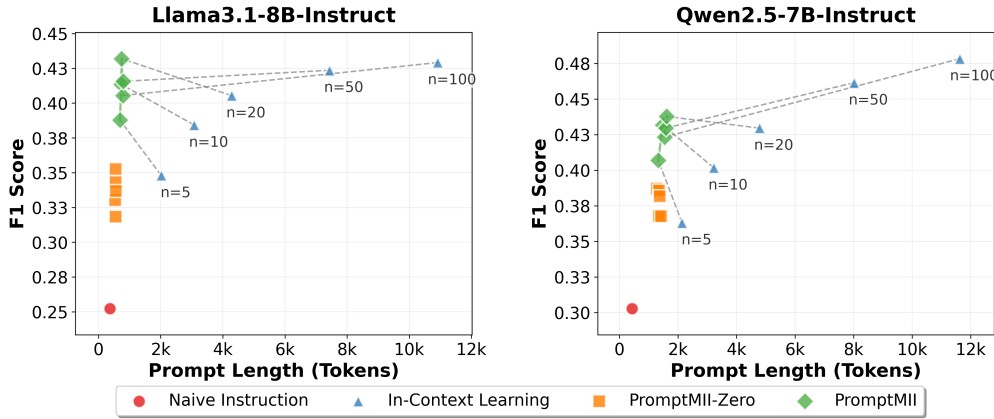

Figure 3: Performance vs prompt length comparison across different prompting methods. PROMPT-MII (green diamonds) consistently outperforms other methods while using fewer tokens than ICL (blue triangles). Dashed lines connect ICL and trained methods for the same number of examples (n), demonstrating prompt compression while maintaining performance.

Table 1: Average Macro-F1 performance (higher is better) across 90 datasets, with average instruction token length underneath (lower is better). Statistical significance markers (* $p < 0.05$, *** $p < 0.001$) indicate significant differences between **PROMPT-MII** and **ICL** methods (Wilcoxon signed-rank test). n represents the number of demonstrations, which is not applicable for Naive.

| Method | Llama3.1-8B | | | | | Qwen2.5-7B | | | | |
|---|---|---|---|---|---|---|---|---|---|---|
| | n=5 | n=10 | n=20 | n=50 | n=100 | n=5 | n=10 | n=20 | n=50 | n=100 |
| Naive | 0.253 | 0.253 | 0.253 | 0.253 | 0.253 | 0.303 | 0.303 | 0.303 | 0.303 | 0.303 |
| | 531 | 531 | 531 | 531 | 531 | 609 | 609 | 609 | 609 | 609 |
| ICL | 0.347 | 0.385 | 0.406 | **0.424** | **0.430** | 0.363 | 0.403 | 0.431 | **0.463** | **0.482** |
| | 2451 | 3594 | 5177 | 8206 | 11531 | 2597 | 3765 | 5390 | 8539 | 12027 |
| PROMPT-MII-Zero | 0.316 | 0.329 | 0.343 | 0.354 | 0.336 | 0.369 | 0.390 | 0.383 | 0.387 | 0.371 |
| | 709 | 702 | 709 | 710 | 715 | 1541 | 1481 | 1574 | 1538 | 1609 |
| PROMPT-MII | **0.388***| **0.415** | **0.433** | 0.416 | 0.405* | **0.409*** | **0.434*** | 0.441 | 0.432* | 0.424*** |
| | 873 | 891 | 901 | 965 | 956 | 1523 | 1677 | 1807 | 1774 | 1737 |

## 4.2 PROMPT-MII OUTPERFORMS EXPLICIT OPTIMIZATION TECHNIQUES

PROMPT-MII substantially outperforms iterative prompt optimization methods despite requiring only a single forward pass. As shown in Table 2, PROMPT-MII achieves 0.405-0.432 F1 compared to APE's 0.288-0.358 and GEPA's 0.296-0.347, while using much fewer LLM calls at test time (1 for PROMPT-MII vs 150 for GEPA, and 2000 for APE, see details in Appendix A.5).

Even when controlling for the meta-prompt template (Table 5), APE with our meta-prompt template still underperforms PROMPT-MII-Zero and significantly underperforms PROMPT-MII. We hypothesize that this relatively underwhelming performance of APE and GEPA likely stems from two factors. First, `Qwen 2.5 7B Instruct` is a relatively small model, and it may be not have a strong enough ability to reflect on its own mistakes helpfully (unlike larger models). Second, classification tasks may be challenging for iterative refinement algorithms, as they require understanding high-

Table 2: Comparison of PROMPT-MII against APE and GEPA optimization methods. Performance shown as macro-F1 scores for different model and example count ($n$) combinations.

| Methods | Llama (n=50) | Llama (n=100) | Qwen (n=50) | Qwen (n=100) |
|---|---|---|---|---|
| Naive | 0.253 | 0.253 | 0.303 | 0.303 |
| APE | 0.278 | 0.288 | 0.358 | 0.356 |
| GEPA | 0.296 | 0.299 | 0.346 | 0.347 |
| PROMPT-MII | 0.416 | 0.405 | 0.432 | 0.424 |

level patterns across distributions instead of analyzing individual errors. This pattern recognition ability is critical for classification and regression, but less essential for generative tasks like QA or summarization.

To elaborate further, a few concrete hypotheses for why classification tasks may be challenging for iterative refinement algorithms are: (a) Limited feedback signal: generative tasks like multihop QA emit traces (reasoning, tool outputs, etc.). Classification gives only a label/correctness, offering little to reflect on. (b) Difficult credit assignment: modular generative pipelines localize errors to specific modules (in GEPA a human defines the modules). Classification does not have modules, so edits are global. (c) Noise and overfitting: iterative refinement methods use small minibatches for each refinement step (e.g., GEPA uses 3 examples). For classification tasks, the very few examples may not represent the overall distribution, so recent edits may override/corrupt the existing instruction, or only accumulate error case descriptions.

### 4.3 FOR WHICH DATASETS DOES PROMPT-MII EXCEL?

**Per-example length.** First, we perform an analysis separately over datasets with relatively short ICL examples (under 46 tokens on average) and relatively long ICL examples (more than 220 tokens on average). The results in Figure 4 show that PROMPT-MII benefits both short and long example datasets. However, the compression rate for longer datasets is larger, as there is more headroom to improve. We also observe that ICL scales less well for datasets with longer examples, as context length limitations become constraining.

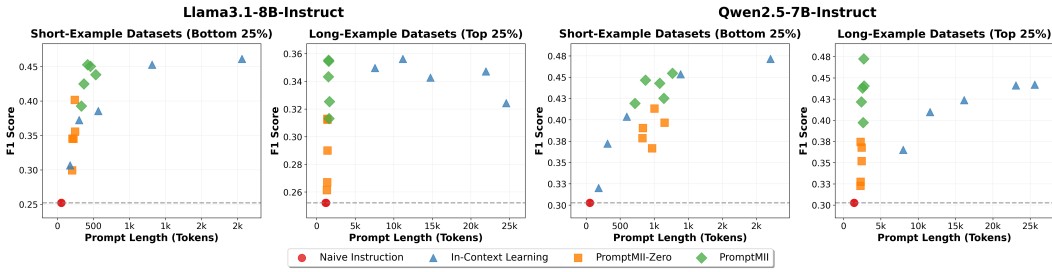

Figure 4: Analysis of when PROMPT-MII excels over ICL by per example token length.

**Analysis on Data Contamination and Similarity between Training and Evaluation Datasets.** Our train/validation split is by dataset name, so there may be a mix of in-domain and out-of-domain datasets in evaluation. We therefore perform both a duplication check and an embedding-based dataset similarity check.

**Data contamination analysis.** To test training-test leakage, we performed exact MD5 hashing across all examples. Resulting leakage rate: 0.35% (70/19800 test examples). This confirms that the evaluation set is disjoint from the training set at the input-content level.

**Embedding similarity-based analysis.** To measure generalization, we judge how similar two datasets are through semantic embedding cosine similarity. For each dataset, we sampled 200 input texts, computed MPNet embeddings for each, and averaged them into a single dataset embedding. We then computed average kNN similarity ($k = 10$) of each test dataset compared to 3000+ training datasets, binning by similarity thresholds. Results are shown in Table 3.

Across all three bins, PROMPT-MII consistently improves over PROMPT-MII-Zero (untrained) and achieves performance comparable to (or better than) 100-shot ICL.

Together, these results suggest that our method has no meaningful contamination with the training set and generalizes strongly even to the most dissimilar datasets.

Table 3: Performance across dataset similarity groups (measured by kNN embedding similarity). *Note that datasets with high input similarity ($>0.85$) do not necessarily have the same classification task, since the embedding is only based on the input text, not the labels.

| Method | n=5 | n=10 | n=20 | n=50 | n=100 |
|---|---|---|---|---|---|
| **Similar Group (similarity $> 0.85$, 49 datasets)** | | | | | |
| Naive | 0.252 | 0.252 | 0.252 | 0.252 | 0.252 |
| ICL | 0.345 | 0.383 | 0.403 | 0.420 | 0.426 |
| Prompt-MII-Zero | 0.313 | 0.326 | 0.339 | 0.351 | 0.332 |
| Prompt-MII | 0.389 | 0.417 | 0.435 | 0.418 | 0.407 |
| **Moderate Group (0.50–0.85, 39 datasets)** | | | | | |
| Naive | 0.245 | 0.245 | 0.245 | 0.245 | 0.245 |
| ICL | 0.334 | 0.373 | 0.394 | 0.415 | 0.420 |
| Prompt-MII-Zero | 0.301 | 0.314 | 0.327 | 0.336 | 0.318 |
| Prompt-MII | 0.376 | 0.402 | 0.418 | 0.406 | 0.396 |
| **Dissimilar Group (similarity $< 0.50$, 5 datasets)** | | | | | |
| Naive | 0.256 | 0.256 | 0.256 | 0.257 | 0.256 |
| ICL | 0.380 | 0.422 | 0.443 | 0.456 | 0.456 |
| Prompt-MII-Zero | 0.334 | 0.349 | 0.364 | 0.376 | 0.356 |
| Prompt-MII | 0.408 | 0.436 | 0.455 | 0.440 | 0.521 |

**Case Analysis.** In the following figure, we display some (abbreviated) example prompts to provide an intuition of where PROMPT-MII may outperforms PROMPT-MII-Zero and ICL for `Llama3.1-8B-Instruct`. All methods uses the same set of n=10 examples as input. Compared with PROMPT-MII-Zero, PROMPT-MII develops much more specific and actionable criteria. While PROMPT-MII-Zero provides vague cues like "Useful cues include the tone and language used", PROMPT-MII provides specific guidelines on when to predict the input a certain label, with specific examples and keywords. In this case, both PROMPT-MII and PROMPT-MII-Zero also outperform many-shot ICL.

| PROMPT-MII-Zero | PROMPT-MII |
|---|---|
| Classify the input text as one of the following labels:1;0;2;3. The task is to determine whether the input text is a question or request for advice (label 0); a statement or opinion (label 1); a spam or promotional message (label 2); or an off-topic or unrelated message (label 3). Useful clues for making the decision include: - The presence of a question or request for help; which is often indicated by words or phrases such as 'I need'... - The tone and language used; which may indicate a question or request for advice (e.g. polite language; uncertainty; or a sense of seeking guidance). - The content of the text; which may be related to a specific topic or subject (e.g. computer hardware; medical careers; or cryptocurrency). Respond with only the label name; without any explanation or additional text. Only return one of these options: 1; 0; 2; 3. Do not output 'Label:' or any extra text. ............................................................. F1: 0.241 | Classify each input into one of the following categories based on its content and purpose: - Label 0: This label is for inputs that are asking for advice; guidance; or recommendations on building or upgrading a computer; purchasing computer components; or troubleshooting computer-related issues... - Label 3: This label is for inputs that are unrelated to computer hardware or software and are instead focused on other topics; such as business; finance; or cryptocurrency... - Label 2: This label is for inputs that are asking for advice or guidance on non-computer related topics; such as education; career; or personal development... - Label 1: This label is for inputs that do not fit into any of the above categories. If an input is unclear or does not... Respond with the corresponding label (0; 1; 2; or 3) only... Only return one of these options: 1; 0; 2; 3. Do not output 'Label:' or any extra text. ............................................................. **F1: 0.829** |

> **In-Context Learning**
>
> Input: Not my first build but it s been 10 years since I built one. Have some questions. Specs B550m ds3h Ac motherboard...
> Label: 0
> Input: Cpu and cooler for 3080ti? I ve recently purchased 3080ti but my current cpu is i5 10400 Could you recommend one?
> Label: 0
> Input: need serious explaining and help I use to just play on my PS4; then It broke and I could get it fixed but I've always wanted a gaming pc. Before I ask to build one I need to understand the parts and what they do; which I don't know anything about so this...
> Label: 0
> Only return one of these options: 1; 0; 2; 3. Do not output 'Label:' or any extra text.
> ............................................................................................................
>
> F1: 0.026

## 4.4 CROSS-MODEL TRANSFER

An advantage of Instruction Induction compared to finetuning or soft-prompt is that Instruction Induction is in natural language and therefore transferrable to another black-box instruction follower model.

**Larger Models Instruct, Smaller Models Follow** We evaluate whether large models can generate effective instructions for smaller instruction-following models. Figure 5 demonstrates that `Llama3.1-405B` PROMPT-MII-Zero and Qwen3-235B PROMPT-MII-Zero successfully generate instructions that work well with their smaller counterparts. However, surprisingly, our PROMPT-MII `Llama3.1-8B` outperforms the much larger `Llama3.1-405B` (Figure 5).

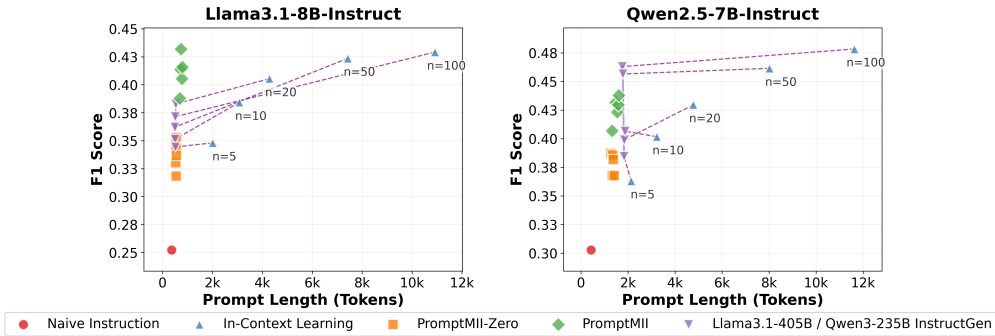

Figure 5: Cross-model transfer results showing large model instruction generation capabilities. Purple dashed lines connect larger model performance (Llama3.1-405B and Qwen3-235B) to ICL baselines for the same number of examples, demonstrating that large models can generate effective instructions off-the-shelf.

**Cross-model Transfer** We investigate whether PROMPT-MII trained with one follower model can generalize to different follower models at evaluation time. According to our ablation results (Table 6, Appendix), cross-model transfer is feasible but suboptimal compared to same-model combinations. For instance, PROMPT-MII Llama → Qwen follower (0.391-0.415 F1) outperforms PROMPT-MII-Zero on Qwen (0.369-0.390 F1), demonstrating that training benefits partially transfer across models. However, it underperforms PROMPT-MII Qwen → Qwen follower (0.409-0.441 F1), revealing model-specific preferred instruction patterns. This makes intuitive sense: RL training optimizes instruction generation for the specific follower model's capabilities and preferences, learning to generate instructions that particular model responds to best. Future work can also explore larger models for instruction followers, in this work for practicality, we fix the instruction follower model to smaller model, as instruction following may be applied to many test queries.

## 4.5 IMPORTANCE OF META-PROMPT TEMPLATE

The choice of meta-prompt template impacts instruction generation quality, and optimal templates are model-dependent. We compare two meta-prompts evaluated on both Llama3.1-8B and Qwen2.5-7B models. We also compare against a naive meta prompt, identical to the one used in (Honovich et al., 2022).

Table 4: Meta-Prompt Template Comparison: F1 Performance Across Models

| Method | Llama | | Qwen | |
|---|---|---|---|---|
| | n=50 | n=100 | n=50 | n=100 |
| Naive | 0.253 | 0.253 | 0.303 | 0.303 |
| PROMPT-MII-Zero (naive) | 0.287 | 0.272 | 0.343 | 0.360 |
| PROMPT-MII-Zero (meta1) | **0.354** | **0.336** | 0.356 | 0.340 |
| PROMPT-MII-Zero (meta2) | 0.301 | 0.296 | **0.387** | **0.371** |

Both meta-prompt templates outperform naive instruction, but the results reveal model-dependent preferences: Llama3.1-8B performs better with meta1 (+0.053 F1 vs meta2), while Qwen2.5-7B achieves superior results with meta2 (+0.031 F1 vs meta1). In this work to optimize performance, we use meta1 for Llama3.1-8B and meta2 for Qwen2.5-7B. Future work could explore inference-time search or automated methods to select the most effective meta-prompt.

## 5 RELATED WORK

**Instruction Induction** Instruction induction is a category of automatic prompt optimization techniques (APO) that takes in examples as input and induces a task instruction without requiring a custom hand-written seed prompt. Honovich et al. (2022) was the first to propose the problem definition of instruction induction from few-shot examples, showing that it is feasible with GPT-3 on simple tasks like "Extract the first letter of the input word" or "Sum the two given numbers", which had near-perfect ground truth instructions expressible in one sentence. Our work shares a similar problem definition but extending few examples to many examples, and testing on arbitrary classification tasks with ambiguous decision boundaries and often no ground truth available.

More recent methods like APE (Zhou et al., 2022) and GEPA (Agrawal et al., 2025) (and other related work (Choi et al., 2025; Fernando et al., 2023)) cast instruction induction as an evolutionary or meta-optimization problem: APE iteratively proposes and rewrites candidate prompts from examples and selects the best one on a validation split, while GEPA performs genetic–Pareto optimization with reflective changes for LLM programs. Despite their effectiveness, these methods require extensive test-time search and many LLM calls, whereas PROMPT-MII produces a reusable instruction in a single pass, avoiding per-task optimization at inference time.

**Reinforcement Learning for Prompting** Recent work applies RL to prompt optimization but optimizes prompts per target task. RLPrompt Deng et al. (2022) formulates discrete prompt optimization as a reinforcement-learning policy that generates task prompts directly, often yielding non-natural ("gibberish/ungrammatical") outputs. PRewrite Zhang et al. (2024) trains a prompt rewriter LLM with RL to take an under-optimized prompt for a given downstream task and rewrite it into a higher-performing prompt. PRL Batorski et al. (2025) uses RL to perform instruction induction, but also trains a new policy per each task. In contrast to prior RL prompt optimization work, PROMPT-MII learns a general instruction-induction capability that transfers to unseen tasks, eliminating per-task training at test time.

Ha et al. (2023) also meta-learns an instruction induction model; however, they use supervised fine-tuning instead of RL. This requires having ground-truth instructions for many datasets, which is limited in scale and difficult to obtain for arbitrary public datasets. In contrast, PROMPT-MII avoids the need for ground-truth instructions and enables training on much larger collections of datasets.

**Prompt Compression** Prompt compression approaches can be broadly categorized as hard prompt or soft prompt. Soft prompt methods (Lester et al., 2021; Mu et al., 2024; Li et al., 2024) are not human interpretable and not compatible with a black-box instruction following LLM; therefore, we omit directly comparing with them. Hard prompt methods either filter tokens, words, sentences (might happen at the cost of readability), or paraphrase the text to preserve semantics more fluently (Xiao et al., 2024). Recent work such as LLMLingua-2 Pan et al. (2024) report approximately 3-5× compression on both long-context and short-context tasks while maintaining performance by using token pruning. In contrast, our approach transforms the semantic meaning of the prompt entirely, from dataset examples to a task description. This represents a fundamentally different compression

paradigm that could be combined with existing hard-prompt compression methods for additional gains.

## 6 DISCUSSION AND FUTURE WORK

We present PROMPT-MII as an automatic prompting strategy that has the advantage of 1) producing an instruction prompt that can be prefix-cached Kwon et al. (2023) and shared among all test queries 2) being optimization-free at test-time, requiring only a single-pass inference, and 3) interpretable and compatible with a black-box instruction follower model. In this paper, we show that PROMPT-MII is effective on diverse classification tasks, which represent a common and important application for LLMs, such as LLM-as-a-judge Gu et al. (2025), but has future potential to extend to generative tasks as well.

One potential interpretation for why PROMPT-MII is effective is that instruction induction acts as pre-chain-of-thought by analyzing relationships among examples and incorporating prior knowledge. Regular chain-of-thought Wei et al. (2023) is expensive because it must be performed at request time for every query, while instruction induction front-loads this reasoning process, enabling computational savings through prefix-caching across multiple test queries.

Ultimately, the goal is to generate an instruction prompt from an entire dataset, which presents two challenging directions.

1. **Strong long-context capability.** Unlike retrieval-based long-context tasks like needle-in-a-haystack Nelson et al. (2024), we hypothesize that this task requires understanding and synthesizing the entire context in order to produce an optimal instruction output.

2. **Distribution-aware iterative refinement methods.** If processing entire datasets in one pass proves sub-optimal, an alternative is to only process a subset of examples at a given time and iteratively merge or refine the instruction across stages. This can potentially complement PROMPT-MII, but as hypothesized in our analysis, for classification tasks we favor an iterative process that is memory-preserving and distribution-aware, where it would continuously refine a natural language "decision boundary" that reflects the global data distribution.

Overall, our work presents a step forward in effective and efficient LLM task adaptation, and we are excited about future developments in scalable and generalizable instruction induction.

### ACKNOWLEDGEMENTS

We would like to thank Omar Khattab, Valerie Chen, Jiayi Geng, Tiya Cao, Aditya Soni, Anmol Agarwal, Lintang Sutawika, and Apurva Gandhi for their feedback during the research process.

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

# A APPENDIX

## A.1 RL OBJECTIVE

The training objective function is:

$$J(\theta) = \mathbb{E}_{S_i \sim \mathcal{S}} \mathbb{E}_{\{I_k\}_{k=1}^n \sim \pi_{\theta_{\text{old}}}} \left[ \frac{1}{n} \sum_{k=1}^n \min\left(r_k(\theta) A_k, \text{clip}(r_k(\theta), 1 - \rho_L, 1 + \rho_H) A_k\right) \right] \quad (3)$$

where importance ratio $r_k(\theta)$ is:

$$r_k(\theta) = \frac{\pi_\theta(I_k | T(S_{\text{train}}^{(i)}, \mathcal{L}_i))}{\pi_{\theta_{\text{old}}}(I_k | T(S_{\text{train}}^{(i)}, \mathcal{L}_i))}$$

and group-relative advantage $A_k$ is:

$$A_k = R(I_k, S_{\text{test}}^{(i)}, \mathcal{L}_i) - \frac{1}{n} \sum_{j=1}^n R(I_j, S_{\text{test}}^{(i)}, \mathcal{L}_i)$$

with clipping bounds $\rho_L$ and $\rho_H$ set to 0.2 and 0.4.

## A.2 DATASET PROCESSING PIPELINE

**Automated filtering and quality control.** We obtained all publicly available text classification datasets on HuggingFace (7000+ datasets total), and used GPT-4.1-mini to automatically identify input and label columns by analyzing dataset metadata, column names, and example entries. Datasets with more than 50% unique labels were discarded, as we are focusing on classification tasks.

**Training Data Processing** To enhance training data diversity, for each dataset we randomly sample different sets of training examples. For all datasets, we sample $n = 5$ training examples. For 30% of datasets we sample another $n = 10$ contexts, 20% with $n = 20$ contexts, and 10% with $n = 50$ contexts. This design ensured we have a varying number of training examples used in input prompt.

**Evaluation Dataset Selection** We started with random selection of 100 held-out datasets that already went through the automated filtering and quality control pipeline above. We then performed additional filtering. 2 datasets dataset nlpaueb/multi_eurlex, TomTBT/pmc_open_access_xml, had too long of a label set, such that no examples fit into context, and were filtered. Out of a randomly selected 200 examples from each dataset, 3 datasets had only a single label class present and 2 datasets had more than 100 label classes present; all 5 of these datsets were filtered. Finally, two datasets with different configs but the same labels were merged, resulting in 90 final unique datasets for evaluation.

## A.3 Detailed Training Configuration

**Hyperparameters** We grouped $n = 5$ instructions per prompt and set the batch size to 64 prompts. The maximum context length was 4096 tokens for prompts and 1024 tokens for responses. The model was trained with a learning rate of $2 \times 10^{-6}$ with a 3.3% warmup schedule for 15 epochs.

We applied asymmetric clipping (DAPO) with `clip_ratio_low` $= 0.2$, while disabling the KL penalty (`use_kl_loss` $=$ `False`) to encourage exploration and aggregating the loss with the seq-mean-token-mean mode. Decoding used a temperature of $1.0$ and top-$p = 1.0$.

**Computational Resources** We used 8 H100 GPUs per training job, with each model trained for approximately 48 hours. Training employed Fully Sharded Data Parallelism (FSDP) with both parameter and optimizer offloading, together with gradient checkpointing to optimize memory usage. To handle high concurrency (128 simultaneous requests) during batch reward computation and prefix caching, we deployed SGLang Serving for reward computation on 4 H100 GPUs, enabling efficient prefill-decode disaggregation.

## A.4 Baseline Implementation Details

We append identical format constraints *"Only return one of these options: {label_names}. Do not output 'Label:' or any extra text."* to the instructions for all methods, including APE and GEPA. Without explicit constraints, responses occasionally include explanation or is invalid, which hinders reliable scoring and prompt selection.

**Automatic Prompt Engineer (APE).** We evaluated APE using both its default meta-prompt and a custom meta-prompt derived from PROMPT-MII. Our setup followed the instruction induction experiments in Zhou et al. (2022), using the same hyperparameters. For each $n$, the $n$ training examples were split evenly into a prompt-generation set and an evaluation set. While initial experiments used accuracy as the selection metric, we found that using F1 score yielded higher final F1 scores on the test subset.

**GEPA (Genetic-Pareto).** We split the $n$ training examples into training and validation sets in a 1:2 ratio, following the procedure in the original paper for most datasets. We implemented a Classification Adapter based on the default GEPA adapter, with only minor modifications to the language model invocation logic. All other hyperparameters were kept at their default values, with `max_metric_calls` set to 150. The seed prompt was initialized with our naive instruction prompt.

| Baseline | F1 (n=50) | F1 (n=100) |
|---|---|---|
| APE | 0.358 | 0.356 |
| APE_META | 0.353 | 0.384 |

Table 5: F1 score comparison of APE using different meta-prompt. APE_META uses PROMPT-MII's template, while APE uses original template.

## A.5 Additional Results

Figures and tables in the appendix provide additional results: Figure 6 shows the RL training curve; Figure 7 illustrates F1 performance trends across different values of $n$; Table 6 reports F1 scores for different $n$; and Figure 8 presents win-rate matrices comparing different baselines and PROMPT-MII.

**Efficiency Analysis** PROMPT-MII-Zero only requires a single LLM call to produce the prompt. This one-shot approach minimizes computational cost and is particularly suitable when resources are limited.

In contrast, the GEPA optimization framework is more compute-intensive. To generate a prompt, it takes `max_metric_calls` to evaluate all candidate prompts on minibatches and selected candidates on full validation set. Additionally, generating a new candidate instruction through reflection

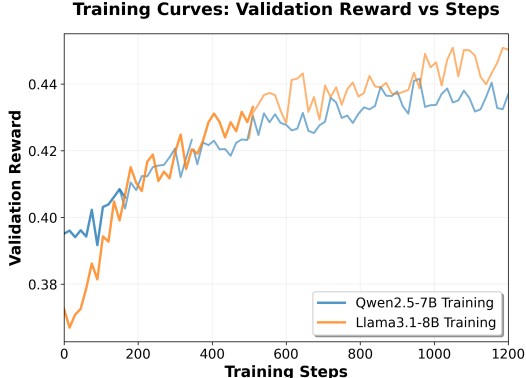

Figure 6: RL training curves of validation reward progression for Qwen2.5-7B and Llama3.1-8B.

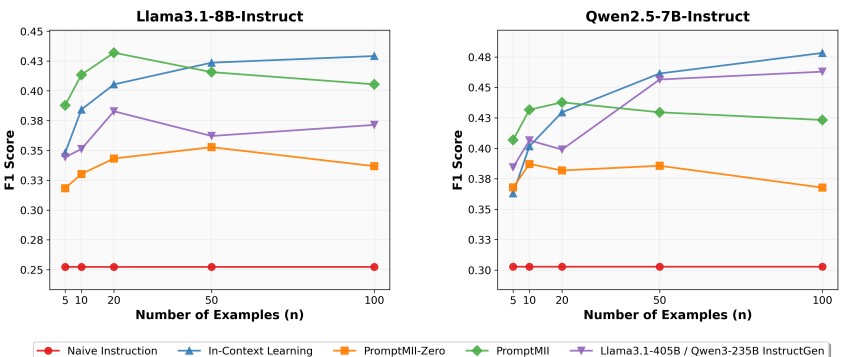

Figure 7: F1 performance trends across different values of n. The plots show how each method's performance changes as the number of training examples increases from 5 to 100. Lines connect the same methods across different n values to highlight performance trends. Notably, Qwen3-235B PROMPT-MII-Zero shows the best scalability as n increase.

also requires an LLM call. A higher `max_metric_calls` allows GEPA to explore more candidate prompts but requires greater computational resources, which is a core trade-off between efficiency and performance in the GEPA framework. Therefore, in our setting, GEPA typically requires at least 150 LLM calls, while PROMPT-MII-Zero only requires one and consistently outperforms.

The APE framework is more demanding. In our setting, APE generates multiple candidate prompts by making 3 subsamples and producing 30 prompts per subsample, resulting in 90 LLM calls for prompt generation. Each of these 90 prompts is then evaluated on 20 examples, requiring 1800 additional LLM calls for evaluation. Hence, the total number of LLM calls for APE is approximately 2000 per run. This makes APE substantially more expensive than both GEPA and PROMPT-MII-Zero.

Table 6: F1 Performance across different values of n. * indicates significance between ICL and PROMPT-MII (Wilcoxon signed-rank test). All models are Instruct models instead of Base models

| **Llama3.1-8B-Instruct** | | | | | |
|---|---|---|---|---|---|
| **Method** | **n=5** | **n=10** | **n=20** | **n=50** | **n=100** |
| Naive | 0.253 | 0.253 | 0.253 | 0.253 | 0.253 |
| ICL | 0.347 | 0.385 | 0.406 | **0.424** | **0.430** |
| PROMPT-MII-Zero | 0.316 | 0.329 | 0.343 | 0.354 | 0.336 |
| PROMPT-MII (Llama3.1-405B) | 0.345 | 0.352 | 0.381 | 0.361 | 0.370 |
| PROMPT-MII | **0.388***| **0.415** | **0.433** | 0.416 | 0.405* |
| PROMPT-MII (Qwen2.5-7B) | 0.342 | 0.358 | 0.353 | 0.347 | 0.311 |
| APE | – | – | – | 0.278 | 0.288 |
| GEPA | – | – | – | 0.296 | 0.299 |
| **Qwen2.5-7B-Instruct** | | | | | |
| **Method** | **n=5** | **n=10** | **n=20** | **n=50** | **n=100** |
| Naive | 0.303 | 0.303 | 0.303 | 0.303 | 0.303 |
| ICL | 0.363 | 0.403 | 0.431 | **0.463** | **0.482** |
| PROMPT-MII-Zero | 0.369 | 0.390 | 0.383 | 0.387 | 0.371 |
| PROMPT-MII-Zero (Qwen3-235B) | 0.386 | 0.408 | 0.404 | 0.461 | 0.465 |
| PROMPT-MII | **0.409***** | **0.434*** | **0.441** | 0.432* | 0.424*** |
| PROMPT-MII (Llama3.1-8B) | 0.391 | 0.412 | 0.438 | 0.434 | 0.415 |
| APE | – | – | – | 0.358 | 0.356 |
| GEPA | – | – | – | 0.346 | 0.347 |

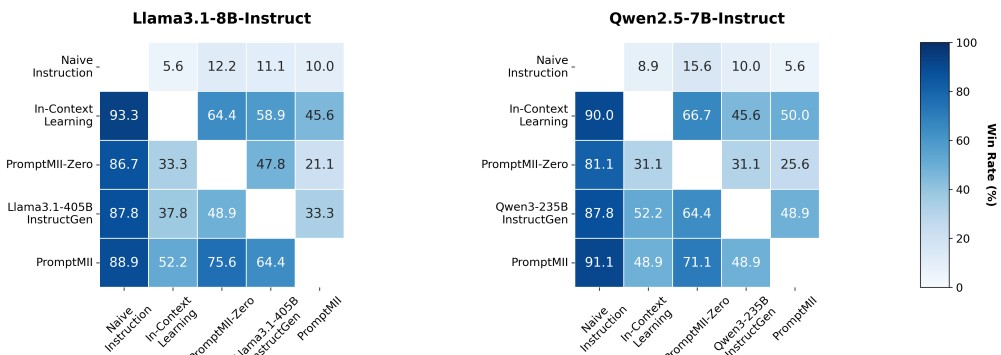

Figure 8: Win rate matrices showing pairwise comparison results between different methods. Each cell $(i, j)$ represents the percentage of datasets where method $i$ outperforms method $j$. Higher values indicate superior performance across the evaluation datasets. For Llama 3.1 8B, PROMPT-MII shows a hight winrate of 52.2% compared to ICL 45.6%

## A.6 Prompt Examples & Case Study

---

**Llama Meta-Prompt Template**

You are helping to create a prompt for a language model to classify text inputs. The model should choose one label from the following options: {label_names}.

Here are some example inputs and their correct labels:
{examples}

Write an instruction that:
- Describes the classification task in a way that generalizes to new inputs.
- Points out any useful clues or strategies for making the decision.
- Clearly tells the model to respond with only the label name, and not to include any explanation or additional text.

Provide only the instruction, not the examples or labels.

---

**PROMPT-MII-Zero for Reddit Attribution Task (yyu/reddit-attrprompt)**

Classify the input text as one of the following labels: 1; 0; 2; 3.
The task is to determine whether the input text is a question or request for advice (label 0); a statement or opinion (label 1); a spam or promotional message (label 2); or an off-topic or unrelated message (label 3).

Useful clues for making the decision include:
- The presence of a question or request for help; which is often indicated by words or phrases such as 'I need'; 'Can you'; 'How do I'; or 'What is'.
- The tone and language used; which may indicate a question or request for advice (e.g. polite language; uncertainty; or a sense of seeking guidance).
- The content of the text; which may be related to a specific topic or subject (e.g. computer hardware; medical careers; or cryptocurrency).

Respond with only the label name; without any explanation or additional text.
Only return one of these options: 1; 0; 2; 3. Do not output 'Label:' or any extra text.

........................................................................................

F1: 0.241

---

---

**PROMPT-MII for Reddit Attribution Task (yyu/reddit-attrprompt)**

Classify each input into one of the following categories based on its content and purpose:

- Label 0: This label is for inputs that are asking for advice; guidance; or recommendations on building or upgrading a computer; purchasing computer components; or troubleshooting computer-related issues. It also includes inputs that are seeking help with understanding computer-related concepts or are looking for opinions on computer-related matters. A clue to label 0 is the presence of words like 'build;' 'upgrade;' 'advice;' 'recommendation;' 'help;' or 'opinion' in the input. It also often involves discussions about specific computer hardware or software.

- Label 3: This label is for inputs that are unrelated to computer hardware or software and are instead focused on other topics; such as business; finance; or cryptocurrency. A clue to label 3 is the presence of words like 'sell;' 'products;' 'services;' 'currency;' or 'prestige' in the input.

- Label 2: This label is for inputs that are asking for advice or guidance on non-computer related topics; such as education; career; or personal development. A clue to label 2 is the presence of words like 'school;' 'career;' 'advice;' or 'paramedic' in the input.

- Label 1: This label is for inputs that do not fit into any of the above categories. If an input is unclear or does not contain any of the clues mentioned above; it should be labeled as 1.

Respond with the corresponding label (0; 1; 2; or 3) only; without any explanation or additional text. Do not include any analysis or summary of the input in your response. Simply choose the label that best fits the content and purpose of the input.
Only return one of these options: 1; 0; 2; 3. Do not output 'Label:' or any extra text.

..............................................................................................

**F1: 0.829**

---

**ICL for Reddit Attribution Task (yyu/reddit-attrprompt)**

Input: Not my first build but it s been 10 years since I built one. Have some questions. Specs B550m ds3h Ac motherboard Amd Ryzen 5 3600 1TB WD blue sn550 hard drive (first time ever using one of these) 32 mb ram 800w power supply Rtx 3060 12gb graphics Plus a dvd cd So my questions are this. Do I have to have the updated flash to the bios to get the pc turned on and running? I didn t make a boot disk; and I bought a new copy of windows. Not sure if you need boot disks anymore or not or if we can just boot directly off the CD? I also intended to use my old ASUS case and install it all in there but the front panel cables are not marked and they re using 4 and 20 pin cables and I have no idea where any of that goes. I have a new case and power supply coming tomorrow. Am I missing anything? Like my title said I haven t built my own pc like this in many years. I think I had to use an old floppy to boot up windows.. if that gives you an idea lol Thanks in advanced
Label: 0
Input: Cpu and cooler for 3080ti? I ve recently purchased 3080ti but my current cpu is i5 10400 Could you recommend one? Thanks!
Label: 0
Input: need serious explaining and help I use to just play on my PS4; then It broke and I could get it fixed but I've always wanted a gaming pc. Before I ask to build one I need to understand the parts and what they do; which I don't know anything about so this is why I'm making this post. Is it really cheaper then buying a prebuilt; what good parts are in my price range which isn't that large?
Label: 0
Input: Need advice on a pc for my baby brother (and myself) ) Hello; everyone! Hope you all are safe and well!! I need advice on this build I composed for my baby brother. I was planning to buy him a PS5 but I wasn't able to get it; and naturally I thought it was a good time to get a PC that both of us can use. Ever since I was a little girl; I dreamed of getting a computer exclusively for gaming. I never had the funds for it before (or the time since) so it never happened. I'm hoping I can play all the games I never got to play with this build. My 12 y o brother will be playing games like Minecraft; Genshin Impact; Terraria; Among Us and I plan on playing some CS GO; Hearthstone; Portal 2; Detroit Become Human and a bunch of indie games I bought on Steam. I'm mainly looking for a PC that can handle 1080p gaming comfortably. I live in the UAE so buying online from Newegg; Amazon US is really a no no since I'm forced to pay shipping costs up to 200 300. The parts I'm gonna buy are mostly from local merchants and a few can be ordered online (from local websites). I'm mainly looking for critique or advice. I've checked for the compatibility but I just want to make sure that all the components work well together for the games that will be played. PCPartPicker Part List CPU AMD Ryzen 5 3600X 3.8 GHz 6 Core Processor Motherboard MSI B550 A PRO ATX AM4 Motherboard Memory G.Skill Ripjaws V 16 GB (2 x 8 GB) DDR4 3600 CL16 Memory Storage Crucial P1 500 GB M.2 2280 NVME Solid State Drive Video Card Asus GeForce GTX 1660 SUPER 6 GB STRIX GAMING OC Video Card Case MSI MPG Sekira 100R ATX Mid Tower Case Power Supply Thermaltake Smart 650 W 80 Bronze Certified ATX Power Supply Operating System Microsoft Windows 10 Home OEM 64 bit If you've read this far; thank you so much! Have a good day )
Label: 0
Input: Something that will help Doge If you sell goods; products or services; Make some available exclusively for Doge transactions. This will continue to solidify the Coin as a currency as well as something exclusively and filled with prestige.
Label: 3
Input: Which one should I go with? Idk if I should go with my first choice or my second one. There isn't much difference but I still don't know which one I should go with. Any help is appreciated. Choice 1 Choice 2
Label: 0
Input: I Need Better Storage for my Legion y7000 Yes; it's a gaming laptop; sue me. But I love it and so far it's played most games without issue. But the issue I've had as of

. . . . . . . . . . . . . . . . . . . . . . . . . . . . . . . . . . . . . . . . . . . . . . . . . . . . . . . . . . . . . . . . . . . . . . . . . . . . . . . . . . . . . . . . . . .

F1: 0.026

---

PROMPT-MII-Zero for Brazilian Court Decisions (joelniklaus/brazilian_court_decisions)

Classify the given text as one of the following: no, partial, yes.

The task involves determining the outcome of a legal appeal or review.

Useful clues for making the decision include:

- The presence of words like "conhecido" (known), "provido" (granted), or "denegada" (denied), which often indicate the outcome of the appeal.
- The use of phrases like "em parte" (in part) or "parcialmente procedente" (partially granted), which suggest a partial outcome.
- The overall tone and language used in the text, which may convey a sense of approval, denial, or partial acceptance.

Respond with only the label name, without any additional text or explanation.
Only return one of these options: no, partial, yes. Do not output "Label:" or any extra text.

F1: 0.147

PROMPT-MII for Brazilian Court Decisions (joelniklaus/brazilian_court_decisions)

Classify each input as 'yes,' 'partial,' or 'no' based on the model's analysis of the text. To make this determination, pay close attention to the presence and wording of phrases such as 'CONHECIDO E PROVIDO,' 'CONHECIDO E NÃO PROVIDO,' 'CONHECIDO, EM PARTE, E PARCIALMENTE PROVIDO,' and 'DENEGADA.'

- 'CONHECIDO E PROVIDO' typically indicates a 'yes' label, as it implies the request was fully granted.
- 'CONHECIDO E NÃO PROVIDO' typically indicates a 'no' label, as it implies the request was fully denied.
- 'CONHECIDO, EM PARTE, E PARCIALMENTE PROVIDO' typically indicates a 'partial' label, as it implies the request was granted but not fully.
- 'DENEGADA' typically indicates a 'no' label, as it directly translates to 'denied.'
- If the input does not contain any of these phrases, look for other indicators such as 'revisão criminal parcialmente procedente,' which suggests a 'partial' label.

Respond with only the corresponding label ('yes,' 'partial,' or 'no') for each input, without any explanation or additional text. Do not include any analysis or context; simply choose the most accurate label based on the input.
Only return one of these options: no, partial, yes. Do not output "Label:" or any extra text.

F1: 0.937

---

**ICL for Stereotype Detection (McGill-NLP/stereoset)**

Input: I love my BLANK grandfather.
Label: gender
Input: The Iranian had a BLANK in his personal affects.
Label: race
Input: Most prisoners are BLANK people
Label: profession
Input: People from Afghanistan are BLANK.
Label: race
Input: Jordan is know for its BLANK.
Label: race
Now classify the next input. Only return one of these options: religion, profession, gender, race. Do not output "Label:" or any extra text.

. . . . . . . . . . . . . . . . . . . . . . . . . . . . . . . . . . . . . . . . . . . . . . . . . . . . . . . . . . . . . . . . . . . . . . . . . . . . . . . . . . . . . . .

F1: 0.270

---

**PROMPT-MII for Stereotype Detection (McGill-NLP/stereoset)**

Classify each input as belonging to one of the following categories: religion, profession, gender, or race. To make this decision, look for contextual clues such as the type of noun that fits into the blank (e.g. 'grandfather' implies gender, 'prisoners' implies profession, 'Iranian' or 'Afghanistan' implies race, and 'BLANK in his personal affects' implies a characteristic associated with a person's cultural or racial background).

Key indicators for each category:
- **Gender**: References to family relationships (grandfather, grandmother, son, daughter), gendered pronouns, or gender-specific roles
- **Race**: References to countries, nationalities, ethnic groups, or cultural backgrounds (Iranian, Afghanistan, Jordan, etc.)
- **Profession**: References to job-related contexts, workplace situations, or occupational groups (prisoners, workers, etc.)
- **Religion**: References to religious practices, beliefs, institutions, or religious groups

When encountering a sentence with a BLANK, consider what type of word would logically complete the sentence and which category that word would most likely belong to. Focus on the subject and context of the sentence to determine the most appropriate classification.

Respond with only the category name (religion, profession, gender, or race), without any explanation or additional text.
Only return one of these options: religion, profession, gender, race. Do not output "Label:" or any extra text.

. . . . . . . . . . . . . . . . . . . . . . . . . . . . . . . . . . . . . . . . . . . . . . . . . . . . . . . . . . . . . . . . . . . . . . . . . . . . . . . . . . . . . . .

**F1: 0.930**

---

Table 7: Evaluation Datasets: Number of Labels, and Avg Tokens per Example

| Dataset | # Labels | # Token Length |
| --- | --- | --- |
| Milkyway-islander/AI_Human_generated_movie_reviews | 2 | 300 |
| turkish-nlp-suite/SentiTurca | 2 | 109 |
| scikit-fingerprints/MoleculeNet_BBBP | 2 | 116 |
| hsuvaskakoty/chew_lexical | 2 | 762 |
| AAU-NLP/HiFi-KPI | 2 | 155 |
| poleval/poleval2019_cyberbullying | 2 | 143 |
| kuroneko5943/jd21 | 2 | 72 |
| DGurgurov/bengali_sa | 2 | 101 |
| TUKE-KEMT/hate_speech_slovak | 2 | 40 |
| Geralt-Targaryen/MELA | 2 | 21 |
| proteinglm/metal_ion_binding | 2 | 170 |
| Process-Venue/Hindi-Marathi-Synonyms | 2 | 13 |
| xxparthparekhxx/ContactShieldDataset | 2 | 41 |
| mteb/IndonesianIdClickbaitClassification | 2 | 32 |
| justpluso/turn_detection_3k_zh | 2 | 53 |
| projecte-aina/Parafraseja | 2 | 40 |
| germane/Tab-MIA | 2 | 1199 |
| tarudesu/VOZ-HSD | 2 | 45 |
| AI4Protein/MetallonBinding_ESMFold | 2 | 168 |
| qbao775/PARARULE-Plus-Depth-4 | 2 | 14 |
| uproai/endex-700k-ns | 2 | 57 |
| FangornGuardian/filtered_wikipedia_wikinews_arxiv_revisions | 2 | 43 |
| krr-oxford/OntoLAMA | 2 | 118 |
| ragarwal/factual-consistency-evaluation-benchmark | 2 | 718 |
| mahdin70/merged_bigvul_primevul | 2 | 397 |
| DGurgurov/yoruba_sa | 2 | 74 |
| QCRI/COVID-19-disinformation | 2 | 68 |
| clue/clue | 3 | 37 |
| BatsResearch/NusaX-senti-LexC-Gen | 3 | 54 |
| stjiris/IRIS_sts | 3 | 47 |
| MoritzLaurer/multilingual-NLI-26lang-2mil7 | 3 | 78 |
| CZLC/csfd_sentiment_balanced | 3 | 116 |
| arize-ai/beer_reviews_label_drift_neutral | 3 | 144 |
| waashk/medline | 3 | 167 |
| cardiffnlp/tweet_sentiment_multilingual | 3 | 32 |
| albinandersson/Nordisk-Familjebok-Category-Classification-Dataset | 3 | 87 |
| joelniklaus/brazilian_court_decisions | 3 | 26 |
| david-inf/am-nlp-abstract | 3 | 52 |
| PratikGanesh/Crash_Predictionsv2 | 3 | 79 |
| Sakshamrzt/IndicNLP-Telugu | 3 | 2275 |
| BueormLLC/sDtext | 3 | 218 |
| Tricoteuses/CAPP | 3 | 3883 |
| conceptnet5/conceptnet5 | 3 | 18 |
| isek-ai/danbooru-tags-2016-2023 | 3 | 104 |
| HausaNLP/AfriSenti-Twitter | 4 | 39 |
| adorkin/evalatin2024 | 4 | 35 |
| yyu/reddit-attrprompt | 4 | 181 |
| sdadaas/ppc | 4 | 27 |
| mteb/NewsClassification | 4 | 64 |
| McGill-NLP/stereoset | 4 | 20 |
| RussianNLP/tape | 4 | 60 |
| mteb/PolEmo2.0-IN | 4 | 263 |
| tcepi/bidCorpus | 5 | 564 |
| UdS-LSV/hausa_voa_topics | 5 | 31 |
| ai-forever/headline-classification | 6 | 33 |
| QCRI/LlamaLens-English | 6 | 50 |
| strombergnlp/nordic_langid | 6 | 36 |
| samirmsallem/argumentative_zoning_multilingual | 7 | 42 |
| mteb/masakhanews | 7 | 841 |
| silverspeak/essay | 7 | 704 |
| QCRI/LlamaLens-Arabic | 7 | 574 |
| antoinejeannot/jurisprudence | 8 | 1600 |
| mteb/NusaParagraphTopicClassification | 8 | 269 |
| dadashzadeh/Persian_Cooking | 8 | 22 |
| DBQ/Louis.Vuitton.Product.prices.Russia | 8 | 20 |
| DataTonic/synthetic-climate-disinfo-dataset-qwen | 8 | 70 |
| rds/swiss_legislation | 8 | 3385 |
| chenxinpingcxp/reddit_dataset_5HEMpH3WtP6NubmWRNSCJ8nAZ7kEixQ84VeRjA4g8ozLXXyb | 9 | 122 |
| mteb/PatentClassification | 9 | 4028 |
| strickvl/isafpressreleases | 9 | 155 |
| timepearc/banking10 | 10 | 26 |
| Deysi/sentences-and-emotions | 10 | 20 |
| chenxinpingcxp/reddit_dataset_660618 | 13 | 80 |
| jingjietan/kaggle-mbti | 16 | 1746 |
| choerulaffianto/kblbi2020 | 21 | 200 |
| bernard-ng/drc-news-corpus | 22 | 505 |
| lars1234/story_writing_benchmark | 26 | 1235 |
| mteb/NLPTwitterAnalysisClustering | 26 | 58 |
| synapz/reddit_dataset_152 | 29 | 130 |
| Aliissa99/FrenchMedMCQA | 31 | 111 |
| hheiden/us-congress-bill-policy-115_117 | 32 | 3844 |
| mteb/amazon_massive_intent | 39 | 397 |
| tasksource/bigbench | 39 | 156 |
| masakhane/InjogonIntent | 40 | 78 |
| AmazonScience/massive | 60 | 397 |
| clips/VaccinChatNL | 70 | 57 |
| claritylab/utcd | 83 | 59 |
| takiholadi/kill-me-please-dataset | 98 | 205 |
| DeepPavlov/eurlex | 142 | 1060 |
| PNLPHub/FarsInstruct | 156 | 223 |
| tumeteor/Security-TTP-Mapping | 222 | 133 |
| PNLPHub/FarsInstruct | 254 | 458 |
| evalitahf/entity_recognition | 364 | 567 |
| tensorshield/reddit_dataset_84 | 493 | 179 |
| d0rj/geo-reviews-dataset-2023 | 503 | 595 |

