# OpenReview forum: "Prompt-MII: Meta-Learning Instruction Induction for LLMs"
_ICLR.cc/2026/Conference — ICLR 2026 Poster_

### Official Review · Reviewer_Fcgk · 2025-10-31

**Soundness:** 2
**Presentation:** 3
**Contribution:** 2
**Rating:** 6
**Confidence:** 4

**Summary:**

This paper trains a language model to generate an instruction given a couple of example input-output pairs of a task via reinforcement learning.  The experiments show that this method achieves the performance of ICL with >20 examples, while consuming much fewer tokens for inference.

**Strengths:**

1. The idea is clearly conveyed.
2. I believe it is the first to use RL to train a general model for instruction induction.
3. The experiment is large-scale, and thus the resulting model might be useful to the community.

**Weaknesses:**

1. The technical contribution / novelty is limited.
2. Missing comparison with a naive meta-prompt.

**Questions:**

1. How do you prevent data contamination? That is, how do you make sure that the training dataset does not contain any questions from the holdout test set?
2. A small issue: Line 125, the hyperlink is wrong.

---

> ### Author Response · Authors · 2025-11-26
>
> **Thank you for your review recognizing the strengths of this paper! We respond to your comments below**
>
> > "The technical contribution / novelty is limited."
>
> We appreciate the reviewer’s acknowledgement under “Strengths” that we are the “first to use RL to train a general model for instruction induction”. We’d like to emphasize that achieving significant gains compared to baselines required non-trivial design choices, including how to construct the training corpus, and a unique reward function that wasn’t explored in prior works. The paper also provides new insights through large scale empirical study.
>
> To respond more thoroughly, we would appreciate it if the reviewer could point out specific prior work they view as having similar findings or methodology, so we can address it directly.
>
> > "Missing comparison with a naive meta-prompt."
>
> Thank you for the suggestion. We would like to emphasize that prompt engineering of the meta-prompt is not our core contribution in this paper, but it could be a useful addition in the ablations section. In response, we conducted additional experiment using the same naive meta-prompt as [1], and here are the results
>
> | Method                                   | Llama n=50 | Llama n=100 | Qwen n=50 | Qwen n=100 |
> |------------------------------------------|------------|-------------|-----------|------------|
> | Naive Instruction                        | 0.253      | 0.253       | 0.303     | 0.303      |
> | **Prompt-MII-Zero (naive meta-prompt)**  | 0.287      | 0.272       | 0.343     | 0.360      |
> | Prompt-MII-Zero (meta1)                  | 0.354      | 0.336       | 0.356     | 0.340      |
> | Prompt-MII-Zero (meta2)                  | 0.301      | 0.296       | 0.387     | 0.371      |
>
>
> [1] Honovich ‘22, Instruction Induction: From Few Examples to Natural Language Task Descriptions
>
> > "How do you prevent data contamination? That is, how do you make sure that the training dataset does not contain any questions from the holdout test set?"
>
> Thank you for raising this point. We verified that training and holdout test set are disjoint.
>
> To do this we performed deduplication across all training and test example questions using MD5 hashing. Among all the test examples, a very small percentage of 0.35% (70/19800) has a near-duplicate in the training set.
>
> > "A small issue: Line 125, the hyperlink is wrong."
>
> Thank you! we will fix it in the paper.

---

### Official Review · Reviewer_dUHf · 2025-11-01

**Soundness:** 3
**Presentation:** 2
**Contribution:** 3
**Rating:** 6
**Confidence:** 3

**Summary:**

LMs has ability to generate better instruction to improve downstream performance. This work proposes to meta-learn this capability across variety of classification datasets, using RL.

**Strengths:**

- Strong motivation. Meta-learning instruction induction to generate optimized instruction for unseen task (instead of direct optimization on it) looks valid and promising direction.
- Experiments are generally well-executed, to test the effectiveness of the method.
- Experiment results support the strength of the method. Prompt-MII vastly outperforms naive instruction, and also is better than the untrained version(”Prompt-MII-Zero”). Prompt-MII matches the performance of ICL with 20-100 demonstrations.

**Weaknesses:**

Authors emphasize meta-learning as a key strength, where trained model can generate better instructions in unseen tasks. However, experiments on OOD generalization is not explicitly conducted. 90 held-out datasets, even though they are not seen during training, may be (or not be) in-distribution to the training set. Additional experiments on OOD generalization, with more explicit control on the domain of test set, will provide more insights on the authors’ proposed method.

**Questions:**

### Questions

1. Could authors discuss this work in relation with below prior works? These works seem to study meta-learning instruction generation. I am open that this work has unique contribution w.r.t. below works, but it will benefit the readers to know how this work differs.
    1. Choi ‘25, System Prompt Optimization with Meta-Learning
    2. Ha ‘23, Meta-Learning of Prompt Generation for Lightweight Prompt Engineering
    on Language-Model-as-a-Service, Findings of EMNLP
    3. Fernando ‘23, Promptbreeder: Self-Referential Self-Improvement Via Prompt Evolution (I see this in bibliography, but cannot find in the main text or appendix)
2. (L250-L253) Could authors elaborate more on why “classification tasks may be challenging for iterative refinement algorithms”, compared to "generative tasks like QA or summarization”? Further explanation will help readers to understand the unique benefit of meta-learning instruction induction over existing non-meta-learning algorithms
3. I have confusion about the naming of Prompt-MII-Zero. It seems to be the “untrained” version of Prompt-MII, which means that simply the LM is prompted to generate an instruction given meta-prompt. Prior works [1, 2] have also shown efficacy of generating instruction using LM, which seems to be largely identical to “Prompt-MII-Zero”. I believe more straightforward naming, or at least more explicit explanation can reduce the confusion of the readers, especially those who are skimming through.
4. (Minor details) L248 only mentions Qwen, but shouldn’t it also include Llama? Or does that explanation only holds for Qwen for some reason? If so, authors should elaborate on that.

[1] Honovich ‘22, Instruction Induction: From Few Examples to Natural Language Task Descriptions
[2] Zhou ‘22, Large Language Models Are Human-Level Prompt Engineers

---

> ### Author Response · Authors · 2025-11-26
>
> **Thank you for your review recognizing the strengths of this paper and providing thoughtful suggestions, we address your questions and comments individually below.**
>
> > "Authors emphasize meta-learning as a key strength, where trained model can generate better instructions in unseen tasks. However, experiments on OOD generalization is not explicitly conducted. 90 held-out datasets, even though they are not seen during training, may be (or not be) in-distribution to the training set. Additional experiments on OOD generalization, with more explicit control on the domain of test set, will provide more insights on the authors’ proposed method."
>
>
> Thank you for the insightful question. We agree that understanding OOD generalization is important. Our current process for splitting the train/test datasets is by dataset name, so it is true that there might be a mix of in-distribution and OOD datasets in the evaluation.
>
> In response, we perform additional analysis using duplication check and embedding based dataset similarity check to provide additional insights.
>
> 1) Data contamination analysis
>
> To test training-test leakage, we performed exact MD5 hashing across all examples.
> Leakage rate: 0.35% (70/19800 test examples). This confirms that the evaluation set is disjoint from the training set on the input content level.
>
> 2) Embedding similarity based analysis
>
> To measure generalization, we judge how similar two datasets are through semantic embedding cosine similarity. For each dataset we sampled 200 input text, computed MPNet embeddings for each, and averaged them into a single dataset embedding. We then computed average KNN similarity (k=10) of each test dataset compared to 3000+ training datasets, binning by similarity thresholds. We group our experiment results by bin, and below are the results for Llama 3.1 8B Instruct.
>
>
> **Similar group (Knn similarity > 0.85, 49 datasets):**
>
> *Note that datasets with high input similarity (>0.85) doesn’t necessarily mean the classification task is the same, since the embedding is only based on the input text, not the labels.
>
>
> | **Method**      | **n=5** | **n=10** | **n=20** | **n=50** | **n=100** |
> | --------------- | ------- | -------- | -------- | -------- | --------- |
> | **Naive**           | 0.252   | 0.252    | 0.252    | 0.252    | 0.252     |
> | **ICL**             | 0.345   | 0.383    | 0.403    | 0.420    | 0.426     |
> | **Prompt-MII-Zero** | 0.313   | 0.326    | 0.339    | 0.351    | 0.332     |
> | **Prompt-MII**      | 0.389   | 0.417    | 0.435    | 0.418    | 0.407     |
>
>
> **Moderate group (Knn similarity 0.5 - 0.85, 39 datasets):**
>
> | Method | n=5 | n=10 | n=20 | n=50 | n=100 |
> |--------|-----|------|------|------|-------|
> | **Naive** | 0.245 | 0.245 | 0.245 | 0.245 | 0.245 |
> | **ICL** | 0.334 | 0.373 | 0.394 | 0.415 | 0.420 |
> | **Prompt-MII-Zero** | 0.301 | 0.314 | 0.327 | 0.336 | 0.318 |
> | **Prompt-MII** | 0.376 | 0.402 | 0.418 | 0.406 | 0.396 |
>
> **Dissimilar group (Knn similarity < 0.5, 5 datasets):**
>
> | Method | n=5 | n=10 | n=20 | n=50 | n=100 |
> |--------|-----|------|------|------|-------|
> | **Naive** | 0.256 | 0.256 | 0.256 | 0.257 | 0.256 |
> | **ICL** | 0.380 | 0.422 | 0.443 | 0.456 | 0.456 |
> | **Prompt-MII-Zero** | 0.334 | 0.349 | 0.364 | 0.376 | 0.356 |
> | **Prompt-MII** | 0.408 | 0.436 | 0.455 | 0.440 | 0.521 |
>
> Across all three bins, Prompt-MII consistently improves over Prompt-MII-Zero (untrained) and achieves performance comparable to/better than 100-shot ICL.
>
> 3. In addition, there is a natural OOD scenario in our setup, which is by per example token length. For training we limit to 4k input token length, so datasets with longer examples are not seen during training. In the paper we show that Prompt-MII is able to generalize to those datasets and have high compression ratios (Figure 4).
>
> Together, these results show that our method
> 1) Has no meaningful contamination with the training set, and
> 2) Generalizes strongly even to the most dissimilar datasets, addressing the reviewer’s concern directly.
>
> We hope this provides useful insights and strengthens the claim that Prompt-MII learns a capability that generalizes beyond tasks seen during training.

---

> > ### Author Response · Authors · 2025-11-26
> >
> > **Here are responses to your questions**
> > > "Could authors discuss this work in relation with below prior works? These works seem to study meta-learning instruction generation. I am open that this work has unique contribution w.r.t. below works, but it will benefit the readers to know how this work differs."
> >
> > Thank you for suggesting these related works, we find them relevant as well and will add citations and a discussion in the related work section. Below we summarize how our approach differs from each:
> >
> > **Choi ‘25, System Prompt Optimization with Meta-Learning**
> >
> > This paper focuses on system prompt optimization, which is orthogonal to task instruction induction. Methodology also differs. This paper uses prompt-based iterative refinement, whereas PromptMii uses RL training. The two directions are complementary, and jointly optimizing system prompts with RL is an interesting future opportunity, inspired by both papers.
> >
> > **Ha ‘23, Meta-Learning of Prompt Generation for Lightweight Prompt Engineering on Language-Model-as-a-Service, Findings of EMNLP**
> >
> > This paper shares the same motivation of instruction induction, but the methodology is different. They use supervised finetuning, and specifically train special padding tokens in between in context examples. Their training data comes from Natural Instructions v2 (NIv2), which has ground truth instructions but they are relatively naive and limited in scale (and it is generally hard to obtain instructions that fully capture the task or dataset distribution, even with human expert labeling).  With RL training in Prompt-MII, the ground truth is not required, allowing much more training data, and the model learns to explore beyond human written instructions.
> >
> > **Fernando ‘23, Promptbreeder: Self-Referential Self-Improvement Via Prompt Evolution**
> >
> > This paper also uses a prompt-based iterative refinement method, in the same category as APE, GEPA, and “System Prompt Optimization with Meta-Learning”
> >
> > > "(L250-L253) Could authors elaborate more on why “classification tasks may be challenging for iterative refinement algorithms”, compared to "generative tasks like QA or summarization”? Further explanation will help readers to understand the unique benefit of meta-learning instruction induction over existing non-meta-learning algorithms"
> >
> > To elaborate, a few potential reasons why  “classification tasks may be challenging for iterative refinement algorithms” are:
> >
> > - Limited feedback signal: generative tasks like multihop QA emit traces (reasoning, tool outputs etc). Classification gives only a label/correctness, offering little to reflect on.
> > - Difficult credit assignment: modular generative pipelines localize errors to specific modules (in GEPA a human defines the modules). Classification doesn’t have modules, so edits are global.
> > - Noise and overfitting: iterative refinement methods use small mini batches for each refinement step (e.g. GEPA uses 3 examples). For classification tasks, the very few examples may not represent the overall distribution, so recent edits may override/corrupt the existing instruction, or only accumulate error case descriptions, which defeats the purpose of trying to compress into a shorter prompt.
> >
> > Even though the current baselines underperform, we see opportunities for iterative refinement to work on classification, especially in conjunction with Prompt-MII, as described in Discussions Section. We consider this a promising direction for future work.
> >
> > > "I have confusion about the naming of Prompt-MII-Zero. It seems to be the “untrained” version of Prompt-MII, which means that simply the LM is prompted to generate an instruction given meta-prompt. Prior works [1, 2] have also shown efficacy of generating instruction using LM, which seems to be largely identical to “Prompt-MII-Zero”. I believe more straightforward naming, or at least more explicit explanation can reduce the confusion of the readers, especially those who are skimming through."
> >
> > Thank you for this feedback, “Prompt-MII-Zero” is indeed the instruction induction method in [1] but with an improved meta-prompt, which is also our “untrained” version.  We will include a more explicit explanation in the revised paper. We also added a direct comparison to [1], with the same meta-prompt,  as per another reviewer's request.
> >
> > [1] Honovich ‘22, Instruction Induction: From Few Examples to Natural Language Task Descriptions
> >
> > > "(Minor details) L248 only mentions Qwen, but shouldn’t it also include Llama? Or does that explanation only holds for Qwen for some reason? If so, authors should elaborate on that."
> >
> > Yes it should also include Llama, we will fix it in the paper, thank you.

---

### Official Review · Reviewer_9dwF · 2025-11-01

**Soundness:** 3
**Presentation:** 3
**Contribution:** 2
**Rating:** 4
**Confidence:** 4

**Summary:**

This paper addresses the challenge of high inference cost in in-context learning, where long context sequences are often required to achieve strong performance on new tasks. To mitigate this issue, the authors propose compressing training demonstrations into a more compact yet informative prompt. They introduce PROMPT-MII, an RL-based framework for meta-learning an instruction induction model capable of generating such prompts. Experimental results on several classification benchmarks demonstrate the effectiveness of the proposed approach.

**Strengths:**

1. The problem of reducing context length in in-context learning is both timely and important, given the growing reliance on in-context learning for various downstream tasks.

2. The method is evaluated on multiple classification tasks, and the results indicate clear improvements over baselines.

**Weaknesses:**

1. The experiments are limited to classification tasks, which makes it difficult to assess the generalizability of the approach. Additional evaluation on generative tasks, such as question answering, would strengthen the contribution.

2. It is unclear how robust the meta-prompt design is across different tasks. The example prompts appear to be tailored to specific classification tasks, raising concerns about adaptability and generality.

**Questions:**

1. How does the proposed method perform on more complex tasks, such as open-ended question answering or reasoning-based generation tasks?

2. When applying the approach to a new domain or task, how should the meta-prompt template be constructed or adapted? More discussion or guidance on this would be helpful.

---

> ### Author Response · Authors · 2025-11-26
>
> **Thank you for your review! We’re glad that you find the paper timely and important, and shows clear improvements over baselines.  We address your comments individually below**
>
> > "The experiments are limited to classification tasks, which makes it difficult to assess the generalizability of the approach. Additional evaluation on generative tasks, such as question answering, would strengthen the contribution."
>
> Thank you for your suggestion. We focused on classification tasks because:
> 1) Text classification is an important use case that already spans a wide range of domains and task types.
>
> In particular, our 90 held-out eval datasets covers toxicity detection, legal, biomedical, topic, news, reviews, hate speech, multilingual tasks, etc, with diverse label count (2-503 classes), and input length per example (20-3883 tokens). We will include the full dataset list with dataset statistics in the revised paper, in addition to releasing the datasets and code.
>
> 2) Classification instruction induction is more challenging than many generative tasks.
>
> The difficulty of instruction induction strongly depends on the task type. For certain generative tasks like GSM8k and Q&A, the output mainly relies on the model knowing prior knowledge/reasoning ability to directly answer the question. In contrast, classification requires the model to infer decision boundaries purely from examples, and this requires stronger information synthesis ability. Empirically, as shown in Section 4.2 of our analysis, baseline iterative refinement methods APE and GEPA struggle on these classification settings, supporting our claim that these tasks are particularly challenging, where Prompt-MII provides a unique solution.
>
> ---
> Regarding your suggestion to extend to generative tasks, it is hard to evaluate generation on the large scale that we evaluated classification on because the evaluation of generated text in general scenarios is not a solved problem (these require different methods such as LLMs as judges with specialized rubrics for each task).
>
> So while using Prompt-MII on generative tasks is possible in theory without changing the algorithm, there are a number of additional details that need to be determined, which is why we limited experiments to classification. We agree that this could be useful future work, and will attempt to do an experiment before the end of the rebuttal period (although experiments take significant time).
>
> > "It is unclear how robust the meta-prompt design is across different tasks. The example prompts appear to be tailored to specific classification tasks, raising concerns about adaptability and generality."
>
> We would like to clarify that the same meta-prompt is applied across all training and test datasets. The meta-prompt design works for text classification tasks in general, and there is no need to tailor it to specific tasks.

---

### Author Response · Authors · 2025-12-04

We thank all three reviewers for their thoughtful and constructive feedback. We appreciate that the reviewers recognize that this paper is a timely solution to an essential problem, and that its large-scale experiments demonstrate clear improvements over baselines.

We have addressed each individual question from all the reviewers in our responses below,
and in the revised PDF we have made the following updates suggested by reviewers, which further strengthens the paper.

* Section 4.2: Added a further explanation of why classification is particularly challenging for iterative refinement baselines, whereas our method shows a clear advantage.
* Section 4.3: Added deeper analysis on generalization, including (1) verification that our test datasets have near-zero leakage with training, and (2) performance breakdown across bins, showing that our method achieves strong improvements even on most dissimilar datasets (OOD datasets).
* Section 4.5: Added a comparison against a naïve meta-prompt for ablations.
* Section 5: Expanded discussion of the related work suggested by reviewers.
* Appendix Table 7: Added the full list of 90 test datasets along with dataset statistics.

---

### Meta-Review · Area_Chair_vuZQ · 2025-12-28

**Summary:**

This paper proposes Prompt-MII, an RL (GRPO)-trained meta-learning framework for instruction induction: given a small labeled dataset, a learned “instruction generator” produces a compact instruction that enables an instruction-following LLM to achieve performance comparable to many-shot ICL with less tokens. The paper trains on 3K+ HuggingFace classification datasets and evaluates on 90 held-out classification tasks, showing consistent gains over naive instructions and untrained induction, and competitive performance vs 20–100 shot ICL with substantial token savings. Reviewers broadly agree the problem is timely and the large-scale experimental effort is strong. The primary remaining limitation is that evidence is classification-only, so the claims should be explicitly scoped to classification instruction induction rather than framed as fully task-general.

**Reviewer Concerns:**

Addressed by rebuttal

1. Data contamination: Authors provided an explicit dedup / hashing-based analysis indicating minimal overlap.
2. OOD generalization: Authors added additional analysis (e.g., similarity breakdowns) suggesting robustness even on more dissimilar held-out datasets.
3. Missing baseline: Authors added comparisons against a naive meta-prompt variant, helping separate improvements from meta-prompt engineering vs RL training.

Still outstanding

1. Scope limited to classification: The decision to evaluate only classification remains a meaningful concern. Even if classification is diverse and arguably challenging (not really convinced me), the paper does not empirically demonstrate transfer to open-ended generation (QA, reasoning, summarization). This doesn’t invalidate the contribution, but it does require more careful and more serious claim-scoping.

Requested change: The authors should explicitly state (in abstract/intro/limitations) that the method is validated for text classification instruction induction, and avoid language implying broad task-general validation unless additional evidence is added.

**Reviewer Scores:**

Reviewer 9dwF: Likely remains cautious due to the lack of generative-task evidence
Reviewer dUHf/Fcgk: Likely increases or solidifies their score given the added OOD/leakage/ablation  analysis and improved positioning/clarity.

---

### Decision · Program_Chairs · 2026-01-26

Accept (Poster)